# The Distribution of Furfuryl Alcohol (FA) Resin in Bamboo Materials after Surface Furfurylation

**DOI:** 10.3390/ma13051157

**Published:** 2020-03-05

**Authors:** Minghui Liu, Wanju Li, Hankun Wang, Xuexia Zhang, Yan Yu

**Affiliations:** 1Institute of New Bamboo and Rattan Based Biomaterials, International Center for Bamboo and Rattan, Beijing 100102, China; lmh1519@163.com; 2SFA and Beijing Co-built Key Lab for Bamboo and Rattan Science & Technology, Beijing 100102, China; 3Guangdong Provincial Key Laboratory of Silviculture, Protection and Utilization, Guangdong Academy of Forestry, Guangzhou 510520, China; liwanju2011@126.com; 4College of Material Engineering, Fujian Agriculture and Forestry University, Fuzhou 350002, China; 13121417614@163.com

**Keywords:** bamboo, furfurylation, nanoindentation, distribution, imaging FT-IR

## Abstract

In this study, bamboo was treated with an optimized surface furfurylation process. With this process, dimensionally stable and highly biologically durable bamboo material could be prepared without mechanical reduction. The anti-swelling efficiency (ASE) could reach 50% with a low weight percent gain (WPG about 13%). By using SEM, nanoindentation, and Imaging FTIR, we demonstrated that this high performance improvement is attributed to the unique furfuryl alcohol (FA) resin distribution pattern in the modified bamboo, namely a higher concentration of FA resin located in the region near to the surface of bamboo, and what is more, the preferred distribution of FA resin within the cell walls of parenchyma cells, which is known to be the weak point of bamboo both for biological durability and mechanical performances. Such graded modified bamboo could be utilized as a reliable engineering material for outdoor applications.

## 1. Introduction

Bamboo, an evergreen plant belonging to the Gramineae family and Bambuseae subfamily, is the second most abundant forest resource with integrated economic, ecological, and social benefits [1,2,3]. It is featured with strong mechanical performances, fast growth, high yield, easy propagation, multiple and wide utilization [4,5,6]. However, bamboo material is susceptible to degradation by fungi mainly due to its high nutritious content compared to wood [7], making it a serious problem for any bamboo enterprise during storage, processing and transportation. Similar to wood material, the hygroscopic nature of cell walls also makes bamboo timber suffer from deformation in response to altering atmospheric humidity [8], especially in terms of the significant differential shrinkage along its radial direction, which originates from the inherent gradient density across the bamboo culm wall. Surface functionalization of biomass plays an important role in the utilization of biomaterials [9,10,11,12]. Some surface functional modifications with nanocoating of ZnO and/or TiO_2_ have been used to improve bamboo performances such as durability [13,14], but the dimensional stability was little or only slightly improved [15,16].

Furfurylation is a promising green and sustainable process that is capable of simultaneously improving the physical, mechanical, and durable properties of wood [17,18]. Furfurylation has been proven to have a markedly positive influence on the dimensional stability and biological resistance of wood [19,20,21,22]. In this process, furfuryl alcohol (FA), a strong polar and low molecular weight organic chemical derived from corncobs or sugar cane residues [23], is impregnated into wood cavities [24]. It is then polymerized in situ under acidic catalysts and an elevated temperature, resulting in a wood–polymer hybrid material with excellent comprehensive performances [25,26]. There is even a possible grafting reaction between FA and the guaiacyl unit of lignin in wood [27,28]. However, the structure of bamboo is different from that of wood, especially at the anatomical level, which makes it necessary to understand the specific distribution of furfuryl alcohol (FA) polymer within the furfurylated bamboo in detail.

To evaluate the distribution of modifiers in wood materials, many methods have been proposed in previously reported articles. SEM and XRD were utilized to investigate bamboo materials modified with graphene oxide, ZnO and the tung oil, respectively [14,29]. Nanoindentation was used to test the effect of adhesive infiltration on the micromechanical properties of S_2_ layer of wood cell walls [30]. imaging Raman spectroscopy was applied to study the interaction of poly(ε-caprolactone) (PCL) with cell wall polymers [8], and imaging Fourier transform infrared spectroscopy microscopy (Imaging FT-IR) was used to track the change in the glucomannan backbone and the C-O groups linked to the aromatic skeleton in the lignin of wood after the compression combined with steam treatment [31]. Many investigations have also been performed on the furfurylation of wood. The distribution of FA resin in wood was observed by SEM and the possible chemical reaction and chemical structure changes during furfurylation were investigated by FT-IR spectra [32,33]. In addition, nanoindentation was used to track changes in micromechanical properties after wood furfurylation [34], which could indicate whether the FA resin has entered the cell wall of the furfurylated wood.

In this study, bamboo was treated with an optimized simple surface furfurylation process explored by our group, which can save energy and achieve low chemical consumption as compared to the traditional vacuum and pressure impregnation method [35]. In addition to testing and evaluating its physical, mechanical, and durability properties, we focused on the multi-scale distribution of furfuryl alcohol (FA) resin in the furfurylated bamboo. SEM was used to describe the FA resin distribution at the tissue scale, and nanoindentation combined with Imaging FT-IR were also applied to explore whether FA resin existed in the cell walls of bamboo after modification, which was widely accepted to play an important role in the performance enhancement of furfurylated bamboo.

## 2. Materials and Methods

### 2.1. Materials

Furfuryl alcohol (CAS: 98-00-0, light yellow liquid, ≥98%), a buffering agent, and catalysts were purchased from Sinopharm Chemical Reagent Co., Ltd. (Shanghai, China). All chemicals used were of analytical grade and all solutions were prepared with deionized water. Moso bamboo (Phyllostachys pubescens Mazei ex H. de Lebaie) strips with regular cross section were purchased from a bamboo flooring enterprise in Zhejiang Province, China. The specimens were prepared from the strips with the number and dimensions shown in Table 1. These specimens were conditioned at 23 °C and 65% relative humidity for at least 30 days before furfurylation.

### 2.2. Preparation of FA Solution

The impregnation solution was prepared according to the procedure described by Li et al. [22]. The solution is composed of 70 wt% FA, 26.25 wt% water, 2 wt% buffering agent, and 1.75 wt% catalyst. Sodium borate was used as a buffering agent, and a composite organic acid composed of oxalic acid and citric acid was used as the catalyst. The proportion between these two types of acids should be carefully controlled to obtain a FA solution with both long pot life and enough catalytic activity for polymerization.

### 2.3. Furfurylation of Bamboo

Before furfurylation, all samples were oven-dried at 103 °C for 20 h, and the oven-dry weight of the samples was measured. Then, the specimens were subjected to a surface furfurylation process consisting of two steps: soaking and curing. In the soaking stage, the specimens were soaked in the FA solutions with 70% concentrations for 36 h under room temperature. The soaked samples were then wrapped in aluminum foils, which serves both to minimize the evaporation of FA and facilitate the penetration of FA into bamboo during the next curing stage. In the curing stage, the FA in the samples were polymerized at 105 °C temperature for 5 h. After curing, the samples were further dried at 60 and 80 °C for 2 h in succession, and later at 103 °C until an oven-dried state was achieved. A detailed description of the experimental workflow is given in Figure 1.

### 2.4. Physical and Mechanical Properties

The weight percent gain (WPG) and equilibrium moisture content (EMC) were tested according to Chinese National Standard GB/T 1928-2009. The modulus of rupture (MOR), modulus of elasticity (MOE) and parallel-to-grain compressive strength (CS) of bamboo were tested according to the Chinese national standard GB/T 15780-1995. Mold resistance was tested by following the method described in the Chinese National Standard (GB/T 18261-2000) using *Aspergillus niger* V.Tiegh, *Penicillam citrinum* Thom, *Trichoderma viride* Pers.ex Fr and *Botryodiplodia theobromae* Pat. Fungal decay resistance was evaluated following the method described in the Chinese National Standard GB/T 13942.1-2009 using white rot fungus *Coriolus versicolor* (CV) and brown rot fungus *Gloeophyllum trabeum* (GT). The resistance of the samples against Formosan subterranean termites (*Coptotermes formosanus*) was carried out in the laboratory using the method specified in the Chinese National Standard GB/T 18260-2000. The specific testing and computing methods were described in detail in the previous article [36].

### 2.5. Nanoindentation

A Triboindenter (Hysitron, Minneapolis, MN, USA) with a Berkovich diamond tip with a radius less than 100 nm was used for indenting. The samples with 5% WPG (treated with 15% FA solution) and 13% WPG (treated with 70% FA solution) were prepared according to the procedure described in Appendix A, and the temperature of the sample chamber were kept at 23 ± 0.5 °C. Three samples were tested at each condition, with valid 15 indents performed per sample.

The procedure started with selecting a target region that is near to the outside region of the sample under an integrated light microscope. A loading scheme with a three-segment load ramp was adopted for the nanoindentation test. The target peak load and the loading–unloading rates were 250 μN and 50 μN/s for all the tests, respectively. After loading for 5 s to attain peak load, the indenter was held at constant load for 6 s and then following an unloading segment for 3 s. The elastic modulus and hardness of materials were calculated using Equations (1) and (2).
(1)$$\frac{1}{E_{r}}=\frac{\sqrt{\pi }}{2\beta }\frac{S}{\sqrt{A_{C}}}$$
(2)$$H=\frac{P_{\max}}{A_{C}}$$
where *E_r_* refers to the reduced modulus, which could be directly used as the elastic modulus of cell walls due to the negligibly difference between *E_r_* and *E* (the common elastic modulus) for soft materials like wood materials [34]. *β* is a numerical factor, and it is assumed to be 1.034 for Berkovich and cube corner. *S* is the unloading stiffness which was determined by the initial slope of the unloading curve. A_*c*_ is the maximum projected contact area, and it can be calculated from the area function of the diamond tip used and the maximum contact indentation depth. *P*_max_ is the load at the maximum indentation depth in one indentation cycle, which could be directly obtained from the instrument.

### 2.6. SEM Observation

For SEM examination, the surface polished specimens treated with 70% FA solution were sputter-coated with an ultra-thin layer of Platinum before imaging. Structural details of the treated and untreated bamboo samples were imaged using a Field Emission Gun Scanning Electron Microscope (ESEM-XL 30, FEI Company, Hillsboro, OR, USA) operated at 7.0 kV.

### 2.7. Imaging FTIR Microscopy

Chemical changes in the bamboo samples after surface furfurylation were characterized by imaging FT-IR microscopy in the mid-IR range. Transverse sections with a thickness of 20 μm were cut from the middle part of the control and two different positions in the middle section of the furfurylated bamboo blocks (Figure 2A). The transmission mode was applied on a Spectrum Spotlight 400 imaging FT-IR system (Perkin Elmer Inc., Shelton, CT, USA), providing a spatial resolution of 6.25 × 6.25 µm. From each specimen, three areas measuring 350 × 350 μm were randomly selected in the transverse section (Figure 2B,D), respectively, using a visible CCD camera. The spectra were recorded with a 6 cm^−1^ spectral resolution between 4000 cm^−1^ and 750 cm^−1^ and a total full-spectral image of each selected region was then obtained (Figure 2C, E). The functions of atmosphere correction, flat correction and baseline offset correction were applied in turn to create corrected spectra. Base-line correction was applied at 1800, 1548, 1189, and 840 cm^−1^. Seven to nine-pixel positions located in the cell walls of both parenchyma cells and fibers in each of the scanning areas were selected for assessing the average spectra of each specific area. The adsorption band at 1508 cm^−1^ belonging to aromatic skeletal vibration of lignin was selected for spectrum normalization based on the following facts. Firstly, our preliminary experiment demonstrated the parenchyma cells and bamboo fibers in the present study had a very close lignin content (22%–24%). Furthermore, lignin was the most stable polymer during furfurylation, whereas both cellulose and hemicellulose might be slightly hydrolyzed by the applied acidic FA solution.

## 3. Results

### 3.1. Physical and Mechanical Properties

Some physical and mechanical properties on the macro scale were collect to prove the success of bamboo surface furfurylation. Table 2 presents the changes in several key physical and mechanical properties of bamboo after the surface furfurylation. The weight percent gain (WPG) of bamboo samples could reach 13.3% after the surface furfurylation, which indicated that the FA polymer was successfully introduced into bamboo structure. The equilibrium moisture content (EMC) of furfurylated bamboo decreased from 9.61% to 6.60% in comparison to the untreated bamboo, corresponding to a 31.32% drop down after modification (Table 2). The reduction in EMC is primarily caused by the introduction of hydrophobic FA resin into the cell walls, which prevents the water molecules from reaching the hydrophilic polymers in the cell walls and decreases the absorption of water molecules in bamboo [10]. Table 2 shows that the furfurylated bamboo has a high Volume ASE (V.ASE) of 51.1%, which indicated that the dimensional stability of the bamboo sample has been greatly improved. Our results demonstrate the feasibility of producing stable bamboo materials under low FA resin introduction. This could be explained by the surface layer shielding effect of the furfurylated bamboo, where the hydrophobic FA resin prevented the water molecules from reaching the hydrophilic polymers in the bamboo cell walls. Furthermore, the possible linkage between the FA molecules and cell wall polymers and the esterification caused by the acidic catalyst will further decrease the absorption of water molecules in bamboo.

Table 2 also provides information on the mechanical properties changes of bamboo upon treatment. The surface furfurylation has not resulted in a significant increase in mechanical properties. The changes in the modulus of rupture (MOR) and the modulus of elasticity (MOE) were similar to the furfurylated wood with the traditional liquid-phase vacuum pressure impregnation (VPI) process, whereas the furfurylated wood with the VPI process normally exhibits a significant increase in CS due to high FA resin filling in the cell cavities, which helps to reinforce the wood structure and leads to enhanced mechanical properties [22,33]. In this study, the WPG of FA resin is too low to improve the compression resistance of bamboo.

### 3.2. The Biological Durability of Surface Furfurylated Bamboo

The biological durability of the furfurylated and control bamboos, which is one of the critical properties of wood or bamboo in building or outdoor application, is compared in Table 3 and Table 4. Furfurylation has been proven as an efficient strategy for improving the biological durability of wood [10,37,38], which was also confirmed in this study. After 30 days of infecting with four fungal, the untreated bamboo exhibited the worst infection value of “4”, while the surface of furfurylated bamboo were not infected at all with “0” infection value (Table 3). The average weight loss ratio of the control and surface furfurylated bamboo is shown in Table 4. For the decay resistance, the unmodified bamboo was seriously decayed, with a large weight loss of 60.48% and 54.36%, respectively, for the white rot fungi and brown rot fungi, while the furfurylated bamboo only lost 7.58% and 8.48% of its initial weight, respectively. In addition, the mean weight loss ratio of the control and furfurylated bamboo samples after 4 weeks’ of termite erosion were 12.8% and 0.82%, respectively, which reveals that the termite resistance of the furfurylated bamboo has been significantly improved. In general, the surface furfurylation was found to be an effective method to obtain bamboo material with excellent resistance to molds, decay, fungi, and termites.

### 3.3. The Distribution of FA Resin in Bamboo

SEM and nanoindentation were used to evaluate the micro and nano-scale distribution of FA resin in bamboo, respectively. Figure 3 shows the cross section of both the furfurylated and control bamboo. There were many starch grains located in the cavity of the parenchymal cells of the control bamboo. After furfurylation, the grains in the cells adjacent to the surface of sample completely disappeared (Figure 3c), and only a small part of them still existed in some parenchymal cells close to the middle part of the furfurylated sample (Figure 3e). The disappearance of starch grains was mainly caused by the hydrolysis of acidic FA solution during the long period of soaking treatment. Figure 3c and e further show that there was no visible FA resin in the parenchyma cell lumens, which indicates that the FA resin might be mainly present in the cell wall of furfurylated bamboo. Compared with the fiber cell walls in the untreated bamboo (Figure 3b), there was obvious layer separation existing in the bamboo fiber cell wall after furfurylation, which is due to the bulking effect of the FA resin in the fiber cell walls (Figure 3d,f). In addition, the damage of the fiber cell walls near the surface of the sample (Figure 3d) is more severe than that of the fiber cell walls inside the sample (Figure 3f). This indicates that the amount of FA resin located in the outer region of the furfurylated bamboo is higher than that in the inner part after the simple soaking impregnation.

In order to determine whether FA can penetrate into bamboo cell walls, the mechanical properties of the fiber cell walls of the control (untreated bamboo) and two kinds of furfurylated bamboo with different WPG (about 5% and 13%, respectively) were measured with nanoindentation (Figure 4). For the furfurylated bamboo, the tested fiber bundles were located at the region close to the surface of the bamboo. The tested areas were also close to the fibers adjacent to the big vessels of vascular bundles, which were expected to have the most abundant FA infiltration. Figure 4a describes the procedure for an in situ imaging nanoindentation test on a bamboo cell wall. The validity of each indentation test could be judged from the position of the indentation in the images before and after indenting. The fiber cell walls’ hardness and indentation modulus of the untreated bamboo and furfurylated bamboo are shown in Figure 4b,c. An increase from the unmodified cell walls to the modified cell walls was observed, and more was observed with the increase in WPG of furfurylated bamboo. For the furfurylated bamboo with 13% WPG, the hardness and indentation modulus of the fiber cell walls show an increase of about 44% and 32%, respectively. The nanoindentation analysis confirms that furfuryl alcohol can penetrate into and polymerize in the cell wall by even a simple soaking impregnation. However, the excess penetration and deposition of FA in bamboo on the contrary will injure the macroscopic mechanical properties of samples, as shown in Table 2. In addition, the hardness increased significantly more than the indentation modulus, which can be ascribed to the lower indentation modulus and the increased hardness of the resin polymer itself [22]. The results indicate that a proper FA solution concentration or WPG should be selected if we want to ensure both high stability and durability, as well as improved mechanical performances.

In addition to nanoindentation analysis, we investigated the chemical changes in furfurylated bamboo cell walls with imaging FT-IR microscopy to explore the different distribution of FA resin in the outer and inner region of treated bamboo. The most valuable characteristic peak of FA resin for analysis in the present study is 1711 cm^−1^, which is assigned to the C=O stretching vibration of γ-diketones formed from the hydrolytic ring opening of the furan rings (Appendix A) [39,40]. In the spectra of the parenchymal cell wall (Figure 5a), the intensity of the band located at 1736 cm^−1^ assigned to the unconjugated C=O stretching vibration of the xylan [31] in bamboo underwent a significant intensity enhancement and slight band shift to 1711 cm^−1^ compared to the untreated samples, which should be due to the superposition effect of the adjacent peak 1711 cm^-1^ from the FA resin. This demonstrated that the FA resin was successfully introduced into bamboo parenchymal cell walls. The peak observed at 1037 cm^−1^ and the small peak at 897 cm^−1^, which were assigned to the stretching vibration of C–O in cellulose and hemicellulose and the β-glyosidic linkages between sugar units, respectively, decreased in the spectra of the parenchymal cell wall of furfurylated bamboo, suggesting the occurrence of acid hydrolysis of hemicelluloses during furfurylation [41]. With a further analysis, it was revealed that the absorption band 1736 cm^−1^ showed a more pronounced broadening in the spectra of the parenchymal cell wall near the surface of the furfurylated bamboo, and even a small peak appeared at 1711 cm^−1^, indicating that the parenchymal cell walls near the surface of the furfurylated bamboo contained more FA resin.

For the bamboo fiber cell walls (Figure 5b), the FTIR absorbance at 1037 cm^−1^ of furfurylated bamboo showed a relatively small decrease compared to the control one, while there was little variation in the 897 cm^−1^ peak. Moreover, the absorption peak at 1736 cm^−1^ showed much smaller variation in intensity, with only a small peak broadening for the fiber cell wall near to the surface of the furfurylated bamboo. These results, different with the parenchymal cell wall, indicate that the acid hydrolysis of hemicelluloses in the fiber cell wall was relatively mild during the surface furfurylation, and the content of the FA resin in the fiber cell wall was very low so that its presence could only be detected in the fiber cell wall near to the surface of the furfurylated bamboo. The present FT-IR spectra indicate that FA resin could enter the cell wall of bamboo through a simple soaking impregnation, and the amount of FA resin located in the outer region is higher than that in the inner part which agrees with the SEM results.

Based on the results of SEM, nanoindentation, and imaging FT-IR, the mechanism of bamboo surface furfurylation is schematically illustrated in Figure 6. During the controlled soaking impregnation, a certain amount of FA monomers penetrated into the porous structure of bamboo due to its strong polarity and small molecular. The following heating of samples that were sealed in the heat-conductive aluminum foils would further push the penetration of FA from the outer region of bamboo to the inner one. A graded distribution of FA resin with a higher concentration in the outer region would then be formed within the modified bamboo, which could increase the bamboo’s height, size, stability, and biological durability while minimizing the negative effect of excess FA in the bamboo on its mechanical properties. Furthermore, the significant differences in the anatomical structure of the two kinds of cells (fibers and parenchyma cells) would produce the differential distribution of FA resin in bamboo. Namely, the parenchyma cells, which have a much larger cavity, larger single pits, and more porosity in cell walls, would accommodate more FA resin, compared to the bamboo fibers. This was clearly demonstrated by the results of the imaging FT-IR measurement. This heterogeneous resin distribution is favorable for bamboo modification as a better modification efficiency could be achieved by more FA resin in the parenchyma cells, the weak point of bamboo both for biological durability and mechanical properties. Due to this unique but suitable FA resin distribution in the furfurylated bamboo, a remarkable performance improvement under a low FA resin introduction could be achieved.

## 4. Conclusions

In this work, the dimensional stability and durability of bamboo were significantly improved by a surface furfurylation with a low WPG via a simple soaking impregnation treatment at room temperature. The SEM results show that no visible FA resin was deposited in the cell cavity. The nanoindentation results demonstrate that FA could enter and polymerize within the bamboo cell walls, while the imaging FT-IR further indicated that the FA resin was mainly located in the parenchyma cells close to the surface area of bamboo. All of the above results demonstrate that the simple soaking impregnation could produce a novelty modified bamboo material with graded and heterogeneous FA resin distribution, which was characterized by minimizing the invalid distribution of FA resin in the inner region of bamboo, while also enhancing the weak tissue parenchyma cells with moderate resin deposition. This unique FA resin distribution pattern could explain the remarkable performance improvement of surface furfurylated bamboo under a low FA resin loading. Such graded modified bamboo could be utilized as a reliable sustainable engineering material for various outdoor applications, such as decking, garden furniture, roof sheathing, and so on.

## Figures and Tables

**Figure 1 materials-13-01157-f001:**
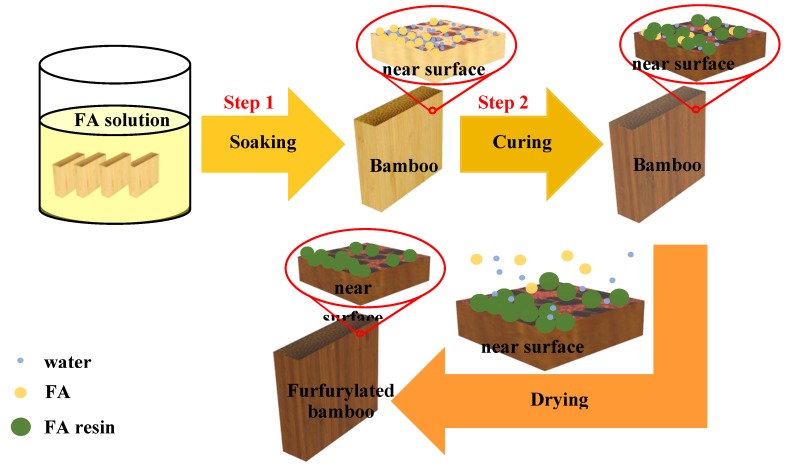
Schematic of wood modification via a simple soaking impregnation with FA: Step1, bamboo samples were soaked in the furfuryl alcohol (FA) solutions for 36 h under room temperature; Step 2, the soaked samples were wrapped in aluminum foils and then cured at 105 °C temperature for 5 h, during this process, FA polymerized into resin while a part of the FA remained unreacted and stored in bamboo; after further drying, the FA resin was further cured and the unreacted FA escaped with the water.

**Figure 2 materials-13-01157-f002:**
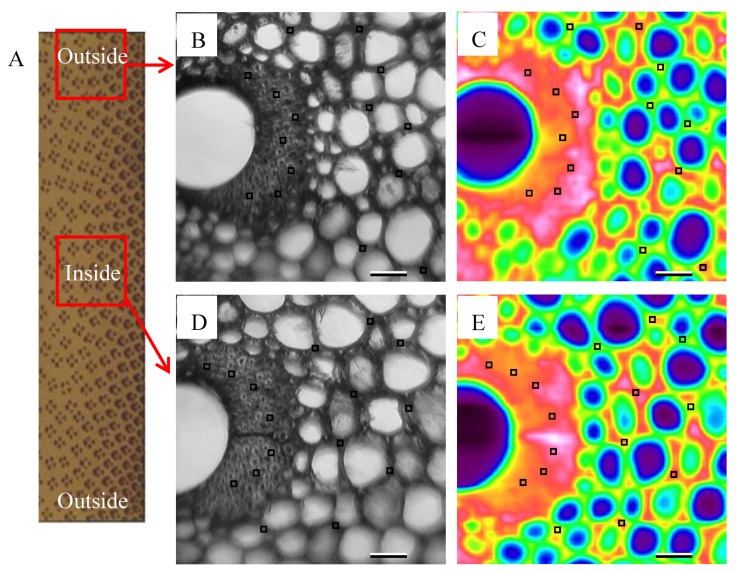
Schematic image of the sampling area for the imaging Fourier transform infrared spectroscopy microscopy (FT-IR) experiment (**A**), and the cross section is in the middle of the furfurylated bamboo samples. Visible-light microscopy image of the test area near the surface of the treated bamboo (**B**), and the test area inside the treated bamboo (**D**). Corrected total full-spectral IR absorbance image of the test area near the surface of the treated bamboo (**C**), and the test area inside the treated bamboo (**E**). The small black empty box indicates the typical pixel position corresponding to the position for the measurement of cell wall spectra. The scale bar = 50 µm.

**Figure 3 materials-13-01157-f003:**
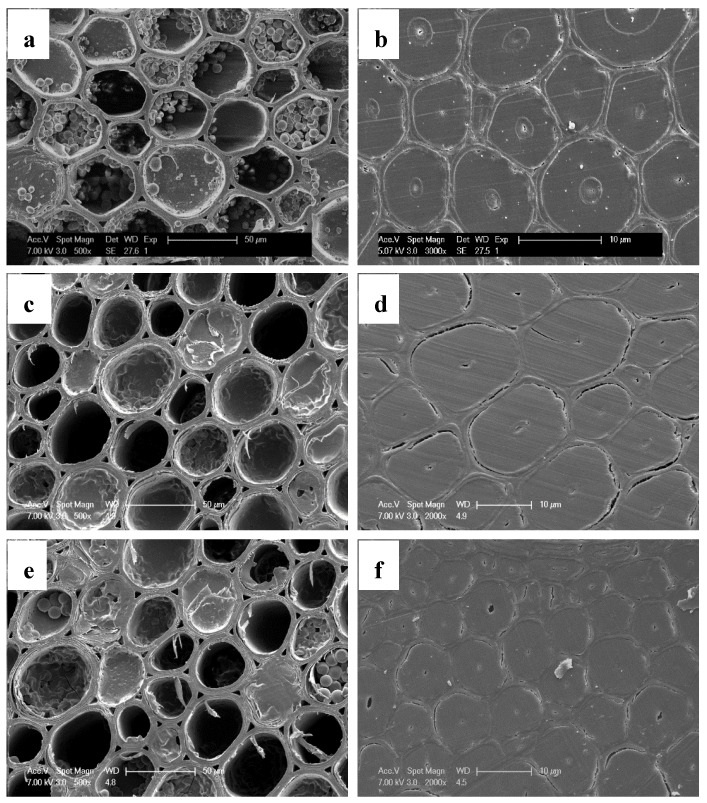
SEM micrographs of microtomed cross-sections: (**a**) the parenchymal cells of untreated bamboo; (**b**) the fibers of untreated bamboo; (**c**) the parenchymal cells near the surface of the furfurylated bamboo; (**d**) the fibers near the surface of the furfurylated bamboo; (**e**) the parenchymal cells close to the center of the furfurylated bamboo; (**f**) the fibers close to the center of the surface of the furfurylated bamboo.

**Figure 4 materials-13-01157-f004:**
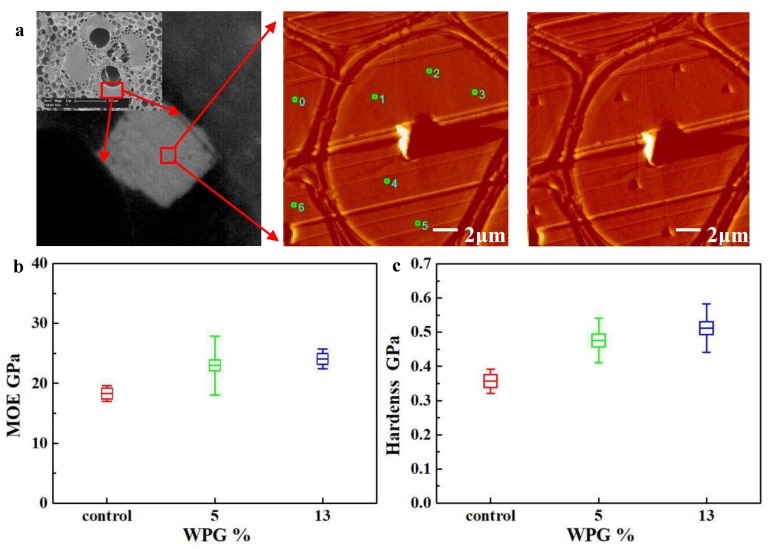
Scanning image of the cell wall before and after indentation (**a**) and the indentation modulus (**b**) and hardness (**c**) of the unmodified and modified fiber cell walls of bamboo.

**Figure 5 materials-13-01157-f005:**
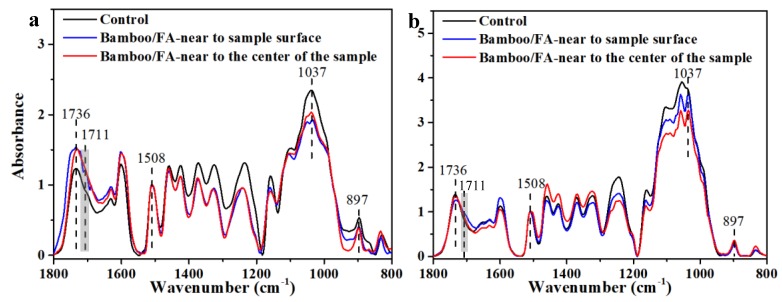
FT-IR spectra of the parenchymal cell walls (**a**) and fiber cell walls (**b**) of the control and surface furfurylated bamboo in the fingerprint region. Black line: FT-IR spectra of the control bamboo; blue-line: FT-IR spectra of the treated bamboo near the sample surface; red line: FT-IR spectra of the treated bamboo near the center of the samples.

**Figure 6 materials-13-01157-f006:**
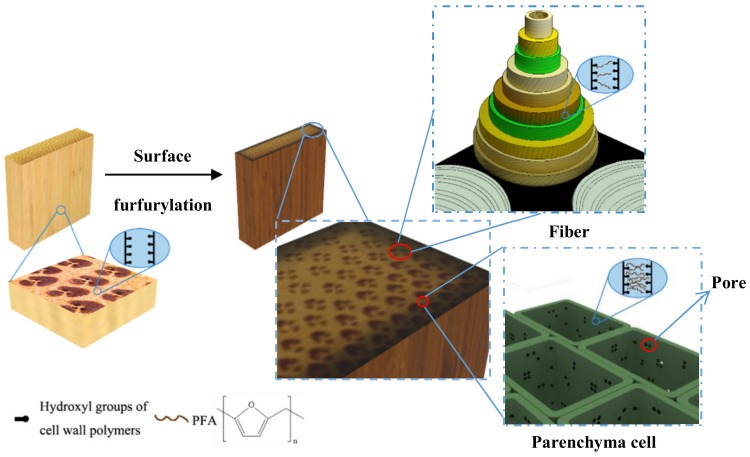
Schematic illustration showing the fabrication process of the surface furfurylated bamboo.

**Table 1 materials-13-01157-t001:** The basic information of specimens.

Items	Size (T × R × L)	Numbers in Each Group
WPG	20 mm × 5 mm × 20 mm	16
EMC and CS	20 mm × 5 mm × 20 mm	15
MOR and MOE	10 mm × 5 mm × 160 mm	11
Mould test	20 mm × 5 mm × 50 mm	24
Decay test	20 mm × 5 mm × 20 mm	16
Termites test	20 mm × 5 mm × 50 mm	5

**Table 2 materials-13-01157-t002:** Physical and mechanical properties of untreated and furfurylated bamboo.

Samples	WPG/%	EMC/%	V.ASE/%	MOR/MPa	MOE/GPa	CS/MPa
Furfurylated bamboo	13.3 ± 2.38	6.60 ± 0.69	51.1 ± 10.4	118.1 ± 20.4	8.91 ± 1.48	52.08 ± 8.58
Control	-	9.61 ± 0.28	-	109.6 ± 12.6	8.72 ± 1.24	51.64 ± 7.66

**Table 3 materials-13-01157-t003:** Results of the mold-resistance test for furfurylated bamboo.

Samples	Infection Value				Resist Effectiveness %
	*A. niger*	*P. citrinum*	*T. viride*	*B. theobromae*	
furfurylated bamboo	0 *	0	0	0	100
control	4 *	4	4	4	0

Note: * Standard method for rating the infection value: 0, the surface of samples has no mycelium; 1, the area of mold infection <25%; 2, the area of mold infection 25%–50%; 3, the area of mold infection 50–75%; 4, the area of mold infection >75%.

**Table 4 materials-13-01157-t004:** The weight loss ratio of furfurylated bamboo after decay and Formosan subterranean termites test.

Samples	Weight Loss Ratio %		
	*Coriolus versicolor*	*Gloeophyllum trabeum*	*Formosan subterranean termites*
furfurylated bamboo	7.85 ± 1.00	8.48 ± 1.29	0.82 ± 0.12
control	60.48 ± 11.00	54.36 ± 12.73	12.80 ± 2.77

Note: The weight loss ratio refers to the ratio between the mass differences of the sample before and after it is corroded by decay or Formosan subterranean termites and the mass of the sample before it is corroded by decay or Formosan subterranean termites.

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
