# Peer review of "The Distribution of Furfuryl Alcohol (FA) Resin in Bamboo Materials after Surface Furfurylation"

_materials, 2020, doi:10.3390/ma13051157_

Round 1

Reviewer 1 Report

Manuscript: The furfuryl alcohol (FA) resin distribution in the  furfurylated bamboo via a simple surface  furfurylation

The manuscript presents very good work related to bamboo surface modification and going to be interesting for the readers.

Some minor comments are as follows;

(1)   Authors need to include some interesting data in the abstract part of the manuscript.

(2)   Why authors used combination of oxalic acid and citric acid was used as the catalyst?

(3)   It will be good if authors can label the corresponding functional group peaks (figure 5).

(4)   Authors need to add future prospective of presented research in the conclusion part of the manuscript.

(5)   Authors need to incorporate some recent reference related to surface functionalization of biomass and their application. For example;

(a) Angew. Chem. Int. Ed. 2005, 44, 3358 – 3393 (b) materialstoday Volume, 21, Issue 7, September 2018, Pages 720-748 (c) Biomacromolecules 18 (8), 2333-2342 (d) ACS Sustainable Chemistry & Engineering 6 (3), 3279-3290 

Author Response

Please see the attachment, thanks for your patience.

Reviewer 2 Report

This article investigated the macro and micro properties of bamboo after modification with furfuryl alcohol. There are some points, which should be considered:

It is always good to see the trend in properties of modified materials. Modification of bamboo with furfuryl alcohol is relatively new, and thus I suggest to include more samples with at least three different WPGs.  

There are some minor grammatical mistakes in the introduction, please revise it.

In introduction, the authors mentioned that there are many studies on the performances of modified bamboo; however, they did not describe the outcome of those researches. For instance, “Nanoindentation was used to investigate the influence of heat treatment on the micromechanical properties of wood-adhesive bonds which was prepared by the phenol formaldehyde (PF) and urea formaldehyde (UF)”, so what was the outcome of this study?

Please describe in MM part 2.1, what are the buffering agent and catalysts?

In part 2.2, please explain the concentration of FA and water.

Have you leach out the unreacted FA? If so, how was the WPG changed?

In nanoindentation, it is mentioned that samples with two different WPGs were tested, while in physical and mechanical properties only the sample with 13.3% WPG were explained.

In MM part, the modification performed with 70% concentration of FA solution, but the obtained WPG was only 13.3%. This WPG is extremely low for such a high concentration of FA. It seems that strong hydrolysis has happened due to the presence of acid catalysts; however, the mechanical results do not show it. Please explain the reasons.

The authors claimed that “Our results demonstrated the feasibility of producing stable bamboo materials under low FA resin loading”; however, in MM part it was mentioned that the concentration of FA is 70%. Please check the text and change the statement.

Please add the standard deviation in table 4.

Author Response

(The authors gave the same response as above.)

Reviewer 3 Report

Definitely, I would change the title of the work as it sounds very redundant. Something more simple and attractive could be adequate.

According to my experience, it is very difficult to use IR spectroscopy (Figure 5) to evaluate the chemical modification of native polymer. It is necessary to have a really high degree of substitution. Moreover, I have evaluated the spectra and the differences are almost imperceptible. Could the authors explain in detail or use another technique for characterization. 

I suggest adding a cartoon for showing the process of chemical modification (furfurylation) of Bamboo.

Do you have an analysis of how the composition of the bamboo is changing after the chemical modification? Could you please comment. 

I think that differential scanning calorimetry (DSC) of modified and raw bamboo could be useful as a part of the characterization. 

Could the authors comment on why they have not used XRD in this work? In line 50 they wrote that method is usually used for they have not explained why they have not used it. Possible advantages of the combination of IR, SEM and nanoindentation tests for the characterization could be highlighted in the conclusions. 

Author Response

(The authors gave the same response as above.)

Reviewer 4 Report

Some aspects should be raised on the manuscript:

Health risks of furfuryl alcohol

Repeated or prolonged contact can cause a skin rash, dryness and redness, see:

Monien, B.H., Herrmann, K., Florian, S., Glatt, H., Metabolic activation of furfuryl alcohol: formation of 2-methylfuranyl DNA adducts in Salmonella typhimurium strains expressing human sulfotransferase 1A1 and in FVB/N mice, Carcinogenesis, Volume 32, Issue 10, October 2011, Pages 1533–1539.

Low cost aspect.

Constructive low cost solutions with bamboo components are of primary relevance to disseminate bamboo technique. See

Sassu, M., De Falco, A., Giresini, L., Puppio, M.L., Structural Solutions for Low-Cost Bamboo Frames: Experimental Tests and Constructive Assessments, Materials 2016, 9(5), 346;

Please furnish some comment on economical feasibility of protection via furfuryl alcohol. Is the durability of furfuryl alcohol verified on full scale bamboo components? Please furnish some detail.

Author Response

(The authors gave the same response as above.)

Round 2

Reviewer 2 Report

Thank you for very good improvement. I have no further comments on your work.

Reviewer 3 Report

You have made a great work with your corrections, but please, only check your title again or discussed with your colleagues.  I suggest using "the furfuryl alcohol (FA) resin distribution of surface furfurylated bamboo".

However, it is up to you how you present your work.